# RayFormer: Improving Query-Based Multi-Camera 3D Object Detection via Ray-Centric Strategies

Xiaomeng Chu
cxmeng@mail.ustc.edu.cn
University of Science and Technology
of China
Hefei, China

Jiajun Deng
jiajun.deng@adelaide.edu.au
The University of Adelaide
Adelaide, Austrilia

Guoliang You
glyou@mail.ustc.edu.cn
University of Science and Technology
of China
Hefei, China

Yifan Duan
dyf0202@mail.ustc.edu.cn
University of Science and Technology
of China
Hefei, China

Yao Li
zkdly@mail.ustc.edu.cn
University of Science and Technology
of China
Hefei, China

Yanyong Zhang*
yanyongz@ustc.edu.cn
University of Science and Technology
of China
Hefei, China

## Abstract

The recent advances in query-based multi-camera 3D object detection are featured by initializing object queries in the 3D space, and then sampling features from perspective-view images to perform multi-round query refinement. In such a framework, query points near the same camera ray are likely to sample similar features from very close pixels, resulting in ambiguous query features and degraded detection accuracy. To this end, we introduce RayFormer, a camera-ray-inspired query-based 3D object detector that aligns the initialization and feature extraction of object queries with the optical characteristics of cameras. Specifically, RayFormer transforms perspective-view image features into bird's eye view (BEV) via the lift-splat-shoot method and segments the BEV map to sectors based on the camera rays. Object queries are uniformly and sparsely initialized along each camera ray, facilitating the projection of different queries onto different areas in the image to extract distinct features. Besides, we leverage the instance information of images to supplement the uniformly initialized object queries by further involving additional queries along the ray from 2D object detection boxes. To extract unique object-level features that cater to distinct queries, we design a ray sampling method that suitably organizes the distribution of feature sampling points on both images and bird's eye view. Extensive experiments are conducted on the nuScenes dataset to validate our proposed ray-inspired model design. The proposed RayFormer achieves 55.5% mAP and 63.3% NDS, respectively.

## CCS Concepts

• **Computing methodologies** → **Object detection**.

*Corresponding author.

## Keywords

3D Object detection, Multiple Camera, Autonomous Driving

**ACM Reference Format:**
Xiaomeng Chu, Jiajun Deng, Guoliang You, Yifan Duan, Yao Li, and Yanyong Zhang. 2024. RayFormer: Improving Query-Based Multi-Camera 3D Object Detection via Ray-Centric Strategies. In *Proceedings of the 32nd ACM International Conference on Multimedia (MM '24), October 28-November 1, 2024, Melbourne, VIC, Australia.* ACM, New York, NY, USA, 10 pages. https://doi.org/10.1145/3664647.3681103

## 1 Introduction

Multi-camera perception, with its omnidirectional view and low-cost deployment, has emerged as a promising solution for autonomous vehicles (AVs). As a critical component of AVs' camera-centric perception systems, multi-camera 3D object detection has garnered significant research interest in recent years [1, 28, 33, 39, 42]. Within the exploration of multi-camera 3D object detectors, query-based approaches [4, 11, 20, 23, 34, 38] achieve inspiring accuracy, securing a pivotal role in contemporary methods.

Typically, a query-based multi-camera 3D object detector begins by initialing object queries as learnable random parameters or as grids uniformly distributed in Cartesian coordinates. The query features are then iteratively refined either by abstracting multi-view image features with a global attention layer or by projecting the 3D query onto the image view to sample the corresponding image features. In this work, we follow the projection and sampling paradigm to perform query feature refinement, as it avoids the large computation overhead introduced by global attention and convergence faster at the training stage.

However, due to the optical characteristics of the camera, 3D positions in an instance frustum tend to project into closely positioned image areas [32, 37]. This presents a dilemma: while we aim for each query to represent an independent object, during the backward projection for feature sampling, multiple independent queries in a grid-like distribution within an instance frustum may project onto the same object, leading to similar features, as depicted in Fig. 1(a). Additionally, the farther away from the camera, the more queries are included within the instance frustum, resulting in a large number of location-distinct queries with semantic-similar features, thus affecting the accuracy of 3D object detection.

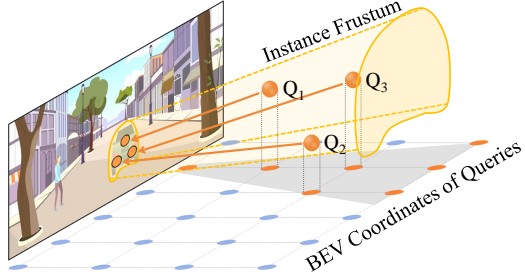

(a) Grid-like query initialization

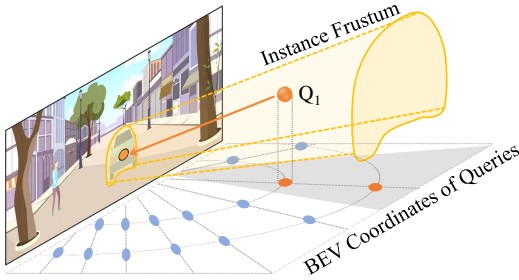

(b) Radial query initialization

**Figure 1: Comparison of query initialization methods: (a) Grid layout results in multiple queries from the instance frustum being projected onto the same object, yielding similar features. (b) Radial initialization mimics optical imaging principles, reducing queries projected onto the same object.**

In this work, we explore how to alleviate the issues outlined above. We find that these problems originate from the practice of grid-like query initialization (*i.e.*, initializing the location of object queries within the Cartesian coordinate system). Such initialization overlooks the actual correspondence between the image and 3D space, where each pixel aligns with a frustum along the camera ray. To this end, we re-formulate the initial location of the object query by well considering the optical characteristics of cameras. As shown in Fig. 1(b), the radial distribution simulates the camera's imaging rays, effectively mitigating the problem of multiple queries projecting onto the same object in the images. Additionally, the number of queries within the frustum does not markedly increase as the distance extends. Moreover, despite that the location of object queries is improved from the radial distribution, queries at different distances along the same camera ray may still sample similar features. To further address this problem, constructing and sampling features from a second perspective, *e.g.*, BEV, help these queries extract distinctive features. Furthermore, each independent query should extract object-level features from a nearby area in the images, thus the sampling points for the query should be placed along the camera ray, aligning with the query's distribution traits.

Formally, we introduce RayFormer, a query-based multi-camera 3D object detection approach that initializes sparse queries in a radial distribution and organizes sampling points along a ray segment for each query to extract both image and BEV features. Specifically, RayFormer leverages image features and predicted depth distribution to generate BEV features via the lift-splat-shoot [31] method.

Using the ego car as the center, the perceptual field is divided by dense camera rays, whereupon the base query points are uniformly and sparsely initiated. Besides, we exploit a 2D prior knowledge to supplement the base queries. This is achieved by categorically setting ray densities and projecting their midpoints onto the images. Subsequently, we select rays intersecting the predicted 2D bounding boxes, procuring foreground queries from these rays. Moreover, the feature sampling scheme should align with the query distribution, ensuring each query extracts object-level features. Given the optical properties of the camera, we no longer select sampling points around queries' locations. Instead, we delineate a segment on the ray at each query's position as sampling units. Our design offers the following benefits. It aligns the initialization with the feature sampling pattern of queries following the optical characteristics of the sensors. Meanwhile, queries along a camera ray embed distinct 3D positional features by sampling the BEV features.

To demonstrate the effectiveness of our proposed RayFormer, we evaluate its performance on the challenging nuScenes benchmark [2]. Without bells and whistles, our approach achieves 55.5% mAP and 63.3% NDS on the test set, which improves the baseline SparseBEV by 1.2% and 0.6%, respectively.

In summary, our main contributions are as follows:

- We propose RayFormer, a novel multi-camera 3D object detection paradigm with a ray-like query initialization and the feature sampling which takes ray segments as units, mirroring the inherent optical characteristics of cameras.
- We construct dense features from both the image and bird's-eye views for feature sampling. Furthermore, we incorporate queries on the foreground rays selected by 2D object detection, without the reliance on inaccurate depth estimations.
- We conduct experiments on the challenging nuScenes dataset. Under the same backbone and training configuration as other studies, RayFormer achieves state-of-the-art performance on the test set, with mAP at 55.5% and NDS at 63.3%.

## 2 Related work

In this section, we give a brief review of polar representation for 3D perception and visual 3D object detection, including BEV-based 3D object detection and query-based 3D object detection.

**BEV-based 3D Object Detection:** BEV-based 3D Object Detection has seen various innovative approaches, mainly categorized by their view transformation techniques—either through forward projection [44, 46], which pioneers the use of the LSS method for feature projection onto BEV, or backward projection techniques [3, 41]. BEVDet [10] is distinguished by its unique data augmentation and refined non-maximum suppression. BEVDepth [16] extends this foundation by incorporating depth estimation supervision and a depth refinement module. BEVFormer [17] further advances the field by dividing the BEV into grids, employing deformable cross-attention [49] to construct BEV features with temporal integration for enhanced detection capabilities [9, 14, 30]. FB-BEV [18] enhances these techniques by introducing a forward-backward view transformation module, while HOP [50] innovates with temporal decoders and object prediction using pseudo BEV features, capturing both spatial and temporal aspects of object motion.

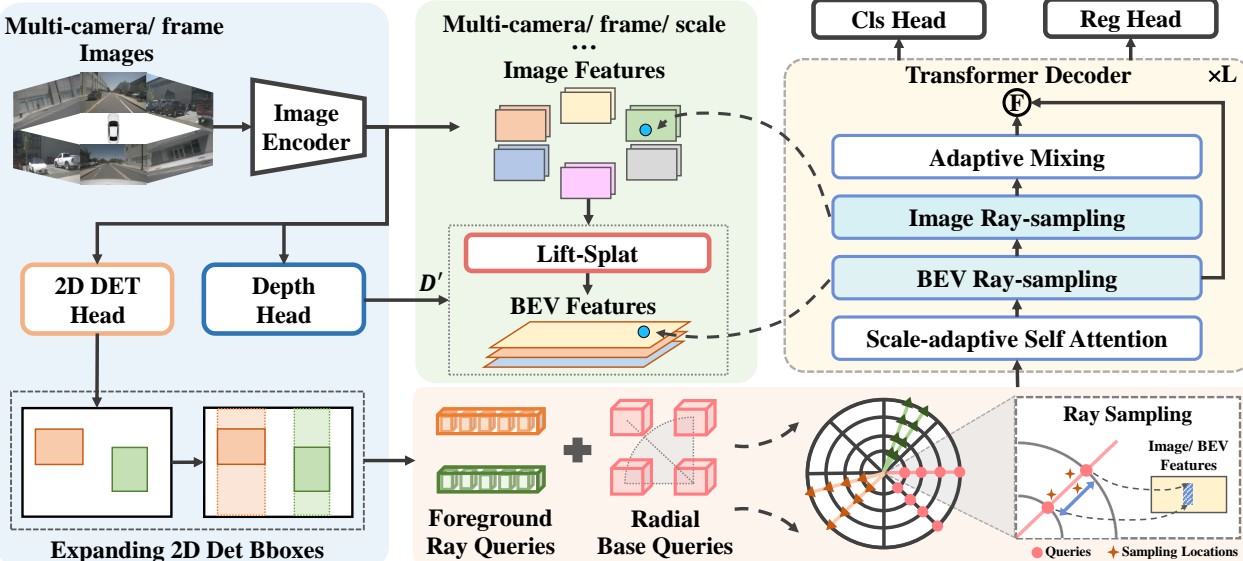

**Figure 2: Overall architecture of RayFormer. Upon inputting multiple frames of multi-camera images into the image encoder, we extract multi-scale image features. These features are processed by a 2D detection head and a depth head to obtain 2D bounding boxes (bboxes) and depth distributions $D'$, respectively. The image features and depth distributions are fed into the lift-splat module for forward projection to generate BEV features. We expand the height of the detected 2D bboxes and use them to select foreground rays. On these rays, a specific number of foreground queries are selected. Along with the radially distributed base queries, all queries are fed into the transformer decoder and refined $L$ times. The core module of the decoder, ray sampling, sets the sampling points along the camera ray, extracting both image and BEV features. Finally, queries are decoded by the classification head and the regression head for accurate predictions.**

**Query-based 3D Object Detection:** DETR3D [38] and PETR [23] utilize transformer decoders to process image features, where the former projects 3D object queries onto 2D features, and the latter integrates the positional embedding of 3D coordinates into image features, yielding 3D position-aware features. StreamPETR [36] builds on PETR with an object-centric temporal mechanism for long-sequence modeling, utilizing historical data through object query propagation. MV2D [40] improves multi-camera detection by generating queries from 2D object detections based on image semantics. Sparse4D [20] meticulously refines anchor boxes through sparse amalgamation and sampling of spatial-temporal features, assigning multiple 4D keypoints to each 3D anchor and fusing these features hierarchically across different views and scales. Sparse-BEV [22] introduces a fully sparse 3D object detection framework that fuses scale-adaptive attention and integrates adaptive spatio-temporal sampling and mixing.

**Polar Representation for 3D Perception:** In the realm of 3D perception, studies leveraging polar representation have shifted from Cartesian to polar coordinates for BEV feature construction. PolarNet [43] pioneers a segmentation approach using a polar BEV representation, optimizing point distribution across grid cells. Polar-BEV [25] enhances processing efficiency through polar embedding decomposition and height-based feature transformation on a hypothetical plane. PolarFormer [12] addresses irregular polar BEV grids and varying object scales with a cross-attention-based detection head and multi-scale learning strategy. TaDe [45] presents a

two-step approach that uses an autoencoder to reconstruct noisy BEV maps, optimizes feature correlations, and converts BEV maps into polar coordinates for enhanced alignment. Contrary to these methods, we construct an implicit query-based polar representation, instead of the BEV feature, through sparse radial query initialization, complemented with a ray sampling method that aligns with the radial query distribution.

## 3  Approach

### 3.1  Overall Framework

As depicted in Fig. 2, RayFormer introduces a novel approach to multi-camera 3D object detection by employing sparse queries to sample features from dual perspectives. The architecture encompasses an image encoder, a 2D detection head, a depth head, and a transformer decoder [5, 35]. The core of the decoder comprises scale-adaptive self attention [22], BEV ray-sampling, image ray-sampling, and adaptive mixing [7], with the add & norm layers and feed-forward network not shown in Fig. 2 for brevity. The process begins with the input of multi-frame, multi-camera images into the image encoder, which consists of a 2D backbone and an FPN, to extract multi-scale image features. These features are then processed by the 2D detection head and depth head to yield 2D bounding boxes and depth distributions. Employing the Lift-Splat technique, we transform the image features and depth distributions from a forward perspective into BEV, resulting in the generation of BEV features. We initiate $N_b$ base queries with a radial distribution. In

parallel, the height of the 2D detection boxes is extended to the full image height, facilitating the selection of corresponding foreground rays on BEV. On these rays, $N_f$ queries are chosen. These $N_b + N_f$ queries are then fed into the transformer decoder. Through scale-adaptive self attention, the queries are encoded with dynamically adjustable receptive fields. Following this, each query, via linear layers, generates multiple sampling points along a specified ray segment, which are then employed to sequentially sample features from BEV and image plane. The image features are aggregated by adaptive mixing and fused with BEV features. In the final step, the refined properties of the queries, as decoded by the classification and regression heads, enable precise object detection.

## 3.2 BEV Feature Generation

We employ the lift-splat-shoot (LSS) method, with depth supervision from the point cloud, as outlined in BEVDepth [16]. This allows us to convert image features into BEV features, presented in Cartesian coordinates. The LSS method innovatively lifts input images into a pseudo point cloud via depth estimation. Subsequently, it aggregates the points within a BEV grid by sum pooling. This procedure is mathematically represented as $BEV(x, y) = \sum_i f_i \cdot \delta(x_i - x, y_i - y)$, where each point $i$ contributes its features $f_i$ to the grid according to its position.

To boost depth estimation precision, we use a linearly increasing discretization method to subdivide the depth range $[d_{min}, d_{max}]$ into varying intervals, each representing a unique category. This approach transforms depth prediction into a more manageable classification challenge within an ordinal regression framework [6], using a bin size of $\sigma$:

$$
\begin{aligned}
\delta &= \frac{2(d_{max} - d_{min})}{K(K+1)}, \\
\hat{l} &= \lfloor -0.5 + 0.5\sqrt{1 + \frac{8(\hat{d} - d_{min})}{\delta}} \rfloor,
\end{aligned}
\tag{1}
$$

where $K$ is the number of depth bins, and $\hat{l}$ is the bin index.

## 3.3 Query Initialization

Owing to the optical properties of the camera, 3D points along a camera ray map onto a closely adjacent area within the image. Centered on the ego vehicle, we use polar coordinates to represent the BEV perception space, that is, first divide $[0, R]$ into $N_r$ segments, where $R$ is the maximum radian value of FoV (field of view), and then divide evenly $[0, D]$ into $N_d$ segments, where $D$ is the maximum value of detection depth range. Consequently, we define the BEV perception area as a circular region with a radius of $D$ and partition the BEV perception field using rays and circles instead of conventional grids. Moreover, the number of queries set along the rays should be small to inhibit the independent queries from sampling similar features. Specifically, we establish a relatively large $N_r$ to densely segment the radian interval, and a relatively small $N_d$ to partition the depth range sparsely. Queries are represented by a combination of C-dimensional features and 3D bounding boxes (bboxes), each of the latter includes four attributes: translation $(r, d, z)$, dimensions $(w, l, h)$, rotation $\alpha$, and velocity $(v_x, v_y)$.

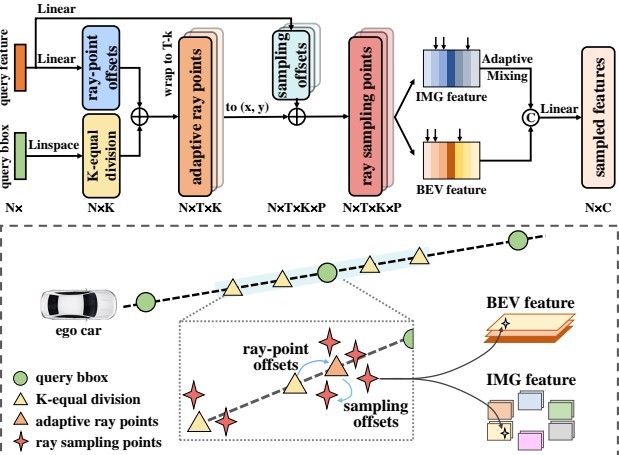

**Figure 3: For $N$ queries, we add $K$ equally spaced points and ray-point offsets to create $N \times K$ adaptive ray points, which are then wrapped to $T$ frames. By incorporating these adaptive ray points and the $P$ sampling offsets generated for each query in Cartesian coordinates, we compile $N \times T \times K \times P$ ray sampling points to aggregate image and BEV features.**

## 3.4 Ray Sampling

For each frame, we uniformly place $N_d$ queries $\{Q_1, Q_2, ..., Q_d\}$ along the rays, where $N_d$ is significantly smaller than the number of rays. Given that we set only a limited number of queries on each ray to extract more comprehensive features of the object, the sampling range for each query is set to be a ray segment. This segment's endpoints are the midpoints between the location of the query and the locations of the two adjacent queries. Along the ray segment, we adaptively select several ray sampling points to extract features from both perspectives, namely the image view and the BEV. In addition to the current timestamp, $T - 1$ historical frames are input to obtain temporal features. We employ a constant velocity model to simulate the movement of objects and dynamically wrap the ray sampling points back to prior timestamps utilizing the velocity vector $[v_x, v_y]$ from the queries [22]:

$$
\begin{aligned}
x_{t,i} &= x_i + v_x \cdot (T_t - T_0), \\
y_{t,i} &= y_i + v_y \cdot (T_t - T_0).
\end{aligned}
\tag{2}
$$

It is worth noting that the polar coordinate system is our default choice. For feature sampling and temporal wrapping, we transform to Cartesian coordinates. Subsequently, queries are converted back to polar coordinates for query refinement.

The specific ray sampling process is illustrated in Fig. 3. For each query, we select $K$ equally spaced points between the endpoints of the defined ray segment and use the query feature to output the offsets for each ray point through a linear layer. By adding the position of K-equal division points and the ray-point offsets, we obtain the adaptive ray points, which are then wrapped to the $T - 1$ previous frames to extract temporal features. Based on the size and orientation attributes of the query bounding boxes, the coordinates of each adaptive ray point are converted into Cartesian coordinates. This conversion facilitates the generation of $P$ sampling offsets

$\{(\Delta x_i, \Delta y_i, \Delta z_i)\}$ through a linear layer, resulting in $N \times T \times K \times P$ ray sampling points. These points are used for sampling image and BEV features, where $N$ represents the number of queries. The sampled multi-frame, multi-point image features are aggregated to $N \times C$ through adaptive mixing. The aggregated image features and BEV features are finally concatenated and fused by a linear layer.

**BEV Ray Sampling:** The historical BEV features with the dimension of $H_B \times W_B \times C$ are wrapped into the ego coordinate system, forming the features $F_B = \{F_t^b | t \in \{1, 2, ..., T\}\}$. We employ deformable attention [48] to sample weighted multi-frame BEV features by $Q_p$. The expression is as follows:

$$BRS(Q_p, F_B) = \frac{1}{T} \sum_{t=1}^{T} w_t \cdot \sum_{j=1}^{N_s} DeformAttn(Q_p, F_{t,j}^b), \quad (3)$$

where $F_{t,j}^b | j \in \{1, 2, ..., N_s\}\}$ denotes the BEV feature point at one of the $N_s$ sampling locations of the t-th frame with the weight $w_t$.

**Image Ray Sampling:** Taking multi-camera images from multiple timestamps as input, the image encoder outputs $T$-frame, $L$-scale features of these $|\Gamma|$-view images. The sampling point $p_j$ along the query $Q_p$'s ray segment projects onto the $|\Gamma|$-view, $L$-scale image features $F^m$ at the t-th timestamp to extract the corresponding pixel's image feature $f_{t,j}$:

$$f_{t,j} = IDA(Q_p, F^m, t, j)$$
$$= \frac{1}{|\Gamma|} \sum_{i \in \Gamma} \sum_{l=1}^{L} w_{i,l} \cdot DeformAttn(Q_p, \mathcal{M}(p_j, i, t), F_{i,l}^m), \quad (4)$$

where $\mathcal{M}$ is the set of camera's projection matrix. $w_{i,l}$ is the weight for the l-th scale in the i-th view.

For each query, we follow the AdaMixer [7] approach by aggregating the $T \times N_s$ feature values through Adaptive Mixing. For the specific implementation, our adaptive mixing is based on Sparse-BEV [22], which elevates the method from 2D to 3D perception. Adaptive mixing comprises two modules: channel mixing and point mixing, which thoroughly decode and aggregate spatio-temporal features across feature channels and point sets.

**Radian Cost for 3D Assigner:** Considering the detailed segmentation of the angular coordinate by rays on the BEV, we aim to primarily refine queries along the depth axis. This effectively redefines the challenge of 3D object localization as a depth estimation task. Therefore, during each transformer layer's query refinement process, we confine the object's positional adjustments within a specified frustum. When aligning predicted 3D objects with their actual counterparts, the angular disparity between normalized i-th predicted radian $\theta_i$ and j-th ground-truth $\theta_j'$ is factored into the matching cost $\Phi$, which is expressed as follows:

$$\Phi_r(i, j) = |(|\theta_i - \theta_j'| + 0.5) \mod 1 - 0.5|,$$
$$\Phi(i, j) = w_c \cdot \Phi_c(i, j) + w_b \cdot \Phi_b(i, j) + w_r \cdot \Phi_r(i, j), \quad (5)$$

where $w_c$, $w_b$ and $w_r$ are the weights of matching cost for classification $\Phi_c$, box attributes $\Phi_b$, and radian $\Phi_r$, respectively.

### 3.5  2D Guided Foreground Query Supplement

In addition to the uniformly distributed base queries described in Sec 3.3, we use 2D object detection to guide the initialization of

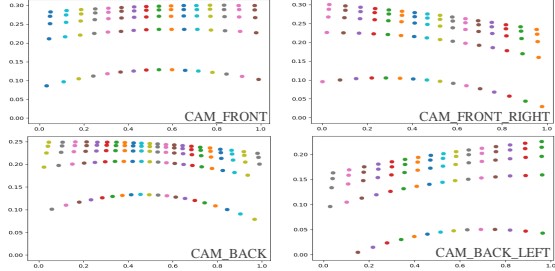

(a) Projection of points on camera rays

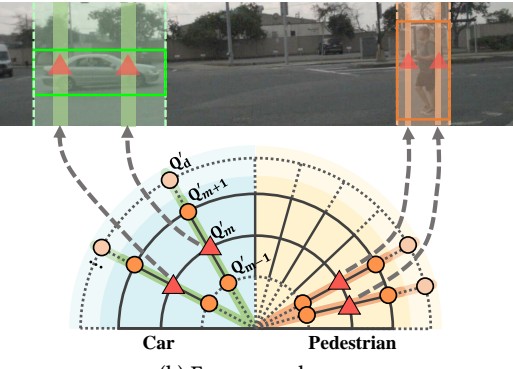

(b) Foreground rays

**Figure 4: The projection of points on camera rays and the selection of foreground rays. (a) Points located on the same camera ray (indicated by same colors) project onto nearly vertical lines within the image. (b) The BEV plane, segmented by rays whose count depends on category sizes, designates rays hitting the expanded areas of category-specific 2D bounding boxes as foreground rays.**

foreground queries, which are then merged with the base queries for prediction. As shown in Fig. 4(a), by utilizing camera intrinsics, we observe that 2D projections from 3D positions on a BEV ray form a near-vertical line, facilitating the detection of 3D objects by checking if any point on the ray aligns with the object's vertical image region. To do this, we extend the detected 2D bounding boxes to the full height of the image to define a broad foreground region. As a result, on the BEV map, we pre-define uniformly distributed rays of varying densities according to the prior dimensions of different categories. We select rays whose midpoints can be projected into the foreground region of each category as the foreground rays. Thus, without relying on the accuracy of depth estimation, the results of 2D object detection alone can guide the initialization of additional queries on these foreground rays. This increases the density of queries around real objects.

Specifically, for category $C$, we divide the BEV map into $N_c$ rays based on its prior dimensions. As shown in Fig. 4(b), in the nuScenes [2] dataset, cars have relatively large prior dimensions, resulting in relatively sparse $N_{car}$ rays, while pedestrians, with relatively small sizes, have a larger number of $N_{ped}$ rays. Next, we project the midpoint $Q_m'$ of each ray onto the image. On each ray whose midpoint falls in the foreground region, we set $N_d'$ queries, which are merged with the base ray queries. Finally, all queries are input into the transformer decoder for subsequent prediction.

**Table 1: Comparison of different methods on the nuScenes val set. † benefits from perspective pretraining. ‡ are trained with CBGS [47] which will elongate 1 epoch into 4.5 epochs.**

| Methods | Input Size | Backbone | Epochs | mAP↑ | NDS↑ | mATE↓ | mASE↓ | mAOE↓ | mAVE↓ | mAAE↓ |
|---|---|---|---|---|---|---|---|---|---|---|
| SOLOFusion [30] | 704×256 | ResNet50 | 90‡ | 0.427 | 0.534 | 0.567 | 0.274 | 0.511 | 0.252 | 0.181 |
| Sparse4Dv2 [21] | 704×256 | ResNet50 | 100 | 0.439 | 0.539 | 0.598 | 0.270 | 0.475 | 0.282 | 0.179 |
| SparseBEV† [22] | 704×256 | ResNet50 | 36 | 0.448 | 0.558 | 0.581 | 0.271 | 0.373 | 0.247 | 0.190 |
| StreamPETR† [36] | 704×256 | ResNet50 | 60 | 0.450 | 0.550 | 0.613 | 0.267 | 0.413 | 0.265 | 0.196 |
| RayFormer † | 704×256 | ResNet50 | 36 | **0.459** | **0.558** | 0.568 | 0.273 | 0.425 | 0.261 | 0.189 |
| BEVFormer† [17] | 1600×900 | ResNet101-DCN | 24 | 0.416 | 0.517 | 0.673 | 0.274 | 0.372 | 0.394 | 0.198 |
| BEVDepth [16] | 512×1408 | ResNet101 | 90‡ | 0.412 | 0.535 | 0.565 | 0.266 | 0.358 | 0.331 | 0.190 |
| SOLOFusion [30] | 512×1408 | ResNet101 | 90‡ | 0.483 | 0.582 | 0.503 | 0.264 | 0.381 | 0.246 | 0.207 |
| SparseBEV† [22] | 512×1408 | ResNet101 | 24 | 0.501 | 0.592 | 0.562 | 0.265 | 0.321 | 0.243 | 0.195 |
| StreamPETR† [36] | 512×1408 | ResNet101 | 60 | 0.504 | 0.592 | 0.569 | 0.262 | 0.315 | 0.257 | 0.199 |
| RayFormer † | 512×1408 | ResNet101 | 24 | **0.511** | **0.594** | 0.565 | 0.265 | 0.331 | 0.255 | 0.200 |

**Table 2: Comparison of different methods on the nuScenes test set. The initialization parameters of VoVNet-99 (V2-99) [13] are all pre-trained from DD3D [29] with extra data. ‡ are trained with CBGS [47] which will elongate 1 epoch into 4.5 epochs.**

| Methods | Input Size | Backbone | Epochs | mAP↑ | NDS↑ | mATE↓ | mASE↓ | mAOE↓ | mAVE↓ | mAAE↓ |
|---|---|---|---|---|---|---|---|---|---|---|
| UVTR [15] | 900×1600 | V2-99 | 24 | 0.472 | 0.551 | 0.577 | 0.253 | 0.391 | 0.508 | 0.123 |
| BEVFormer [17] | 900×1600 | V2-99 | 24 | 0.481 | 0.569 | 0.582 | 0.256 | 0.375 | 0.378 | 0.126 |
| PETRv2 [24] | 640×1600 | V2-99 | 24 | 0.490 | 0.582 | 0.561 | 0.243 | 0.361 | 0.343 | 0.120 |
| PolarFormer [12] | 640×1600 | V2-99 | 24 | 0.493 | 0.572 | 0.556 | 0.256 | 0.364 | 0.439 | 0.127 |
| Sparse4D [20] | 640×1600 | V2-99 | 48 | 0.511 | 0.595 | 0.533 | 0.263 | 0.369 | 0.317 | 0.124 |
| SOLOFusion [30] | 640×1600 | ConvNeXt-B | 90‡ | 0.540 | 0.619 | 0.453 | 0.257 | 0.376 | 0.276 | 0.148 |
| SparseBEV [22] | 640×1600 | V2-99 | 24 | 0.543 | 0.627 | 0.502 | 0.244 | 0.324 | 0.251 | 0.126 |
| RayFormer | 640×1600 | V2-99 | 24 | **0.555** | **0.633** | 0.507 | 0.245 | 0.326 | 0.247 | 0.123 |

## 4 Experiment

### 4.1 Datasets and Metrics

We conducted experiments on the expansive nuScenes autonomous driving dataset [2], a resource rich in perception challenges such as object detection, motion tracking, and LiDAR-based segmentation. This dataset is organized into a total of 1,000 instances, systematically divided across training (700 instances), validation (150 instances), and testing (150 instances). Each instance captures 20 seconds of detailed sensory observations, marked with key-frames twice per second. The sensor suite mounted on the data collection vehicle includes a single LiDAR unit, a hexad of cameras providing complete environmental coverage, and a quintet of radars. For evaluating the precision of object detection, the nuScenes benchmark has established a suite of true positive metrics: Average Translation Error (ATE), Average Scale Error (ASE), Average Orientation Error (AOE), Average Velocity Error (AVE), and Average Attribute Error (AAE), targeting assessment of positional, dimensional, angular, motional, and categorical accuracy, respectively. Furthermore, the Mean Average Precision (mAP) and the holistic nuScenes Detection Score (NDS) serve as key indicators of detection efficacy, with higher values signifying more accurate performance.

### 4.2 Implementation Details

The perception area for BEV is confined to a circle with a 65$m$ radius, segmented by 135 base rays in polar coordinates. On each ray,

which ranges from 0 to 65, we uniformly select 6 queries. Additionally, 30 foreground rays are selected by 2D object detection with 3 points on each ray. This results in a total of 900 queries, which align with previous methods [20, 22, 36]. The architecture of the transformer decoder is designed with 6 layers with shared weights across these layers for efficiency. During ray sampling, we select K points dynamically between midpoints of consecutive queries on the ray. Each ray point, processed through a linear layer, generates 4 sampling offsets. We set K to 5 for BEV feature sampling and 3 for image feature sampling. The Hungarian algorithm is employed to assign predictions to ground truths by utilizing a focal loss [19] as the cost for classification, L1 loss as the cost for both box and angular regression, with the coefficients set at 1, 0.25, and 1, respectively. We generate a depth distribution that is downsampled by a factor of 16 relative to the input. Subsequently, the LSS method is utilized to produce BEV features in Cartesian coordinates. The size of the BEV features is determined by the input image resolution; for a resolution of 1600 × 640, the feature size is 256 × 256, and for other resolutions, it is 128 × 128. In the comparison with other studies, we use T = 8 frames, with an approximate interval of 0.5 seconds between adjacent frames. For ablation studies, we default to conduct experiments with T = 1 frame to validate the function of each module. Futhermore, we train SparseBEV [22] as the baseline under our settings. For a fair comparison, we set the number of sampling points per query point in SparseBEV to 12, aligning it with the total number of sampling points selected along ray segment for

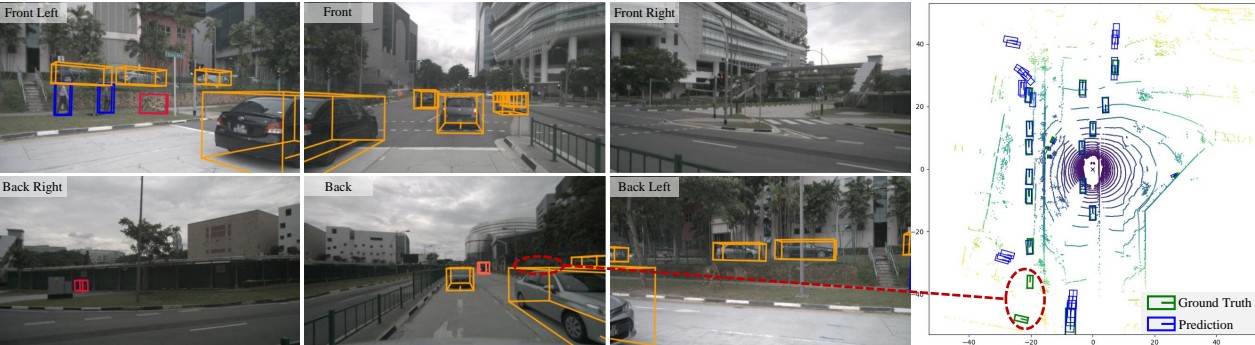

**Figure 5: Visualization of RayFormer. In the BEV diagram (right), ground truth and predicted outcomes are depicted with green and blue rectangles, respectively. Instances of missed detection boxes are highlighted with red circles.**

**Table 3: Comparison of different methods on the nuScenes val set with T=1. All models use ResNet50 as the backbone. † benefits from perspective pretraining. The PETR and BEVDet are trained with CBGS [47].**

| Methods | Input Size | **mAP** | **NDS** | mATE |
|---|---|---|---|---|
| BEVFormer[17] | 704×256 | 0.297 | 0.379 | 0.739 |
| BEVDet[10] | 704×256 | 0.298 | 0.379 | 0.725 |
| PETR[23] | 384×1056 | 0.313 | 0.381 | 0.768 |
| BEVDepth[16] | 704×256 | 0.322 | 0.367 | **0.707** |
| Sparse4D[20] | 704×256 | 0.322 | 0.401 | 0.747 |
| SparseBEV†[22] | 704×256 | 0.329 | 0.416 | - |
| RayFormer † | 704×256 | **0.350** | **0.420** | 0.709 |

**Table 4: Ablation analysis of RayFormer's crucial modules on the nuScenes val dataset. The acronyms RQI, IRS, BRS, 2D AL, and FQS denote radial query initialization, image ray sampling, BEV ray sampling, 2D object detection auxiliary learning, and foreground query supplement, respectively.**

| ID | RQI | IRS | BRS | 2D AL | FQS | mAP↑ | NDS↑ | mATE↓ |
|---|---|---|---|---|---|---|---|---|
| A | | | | | | 0.319 | 0.399 | 0.729 |
| B | ✓ | | | | | 0.316 | 0.399 | 0.740 |
| C | ✓ | ✓ | | | | 0.332 | 0.404 | 0.726 |
| D | ✓ | ✓ | ✓ | | | 0.345 | 0.406 | 0.717 |
| E | ✓ | ✓ | ✓ | ✓ | | 0.347 | 0.414 | 0.716 |
| F | ✓ | ✓ | ✓ | ✓ | ✓ | 0.350 | 0.420 | 0.709 |

each query in our RayFormer. To achieve rapid convergence, we incorporate the query denoising strategy from PETRv2 [24].

RayFormer is trained using the AdamW [27] optimizer with a global batch size of 8. The initial learning rates for the backbone and other parameters are set at 2e-5 and 2e-4, respectively, with a cosine annealing [26] decay strategy. We utilize ResNet [8] and VoVNet-99 (V2-99) [13] as our backbone networks for feature encoding. ResNet parameters are pretrained on nuImages [2], and V2-99 parameters are initialized from DD3D [29]. All models are trained for 24 epochs without the use of CBGS [47] or test time augmentation.

### 4.3 Main Results

**Comparison with the State-of-the-art Methods:** Among all the non-temporal models, RayFormer achieves the highest scores in mAP and NDS on the nuScenes validation set, showing significant improvements of 2.1% and 0.4% respectively over SparseBEV, as illustrated in Tab. 3. The comparison results with multi-frame input (T=8) on the val set are presented in Tab. 1. With an input resolution of 704 × 256 and a ResNet50 backbone, RayFormer reaches 45.9% in mAP and 55.8% in NDS, outperforming StreamPETR by 0.9% and 0.8% respectively. At an input resolution of 512 × 1408 and a ResNet101 backbone, RayFormer scores 51.1% in mAP and 59.4% in NDS, surpassing StreamPETR by 0.7% and 0.2%, respectively. The comparison of RayFormer with other advanced methods on the

nuScenes test set is presented in Tab. 2. For this test, we employ an input resolution of 1600 × 640 and selected VoVNet-99 as the backbone. RayFormer scores 55.5% in mAP and 63.3% in NDS, which surpasses the baseline SparseBEV by 1.2% and 0.6% respectively.

**Visualization Results:** Fig. 5 showcases a system visualization with an input resolution of 512 × 1408, using ResNet101 as the backbone. The BEV layout on the right captures the full scene, with blue and green boxes indicating predictions and ground truths. The left side shows 3D bounding box projections onto the image plane, with yellow, blue, and red coding for cars, pedestrians, and bicycles, respectively. Missed detections are outlined in dashed red, matching the BEV view. Fig. 6 illustrates the ray sampling method, marking high-scoring query sampling points on the BEV plane and images with red dots. These points exhibit a radial pattern on the BEV and a vertical alignment in images, focusing on object areas due to the radial sampling strategy.

### 4.4 Ablation Studies

**Image Ray Sampling:** Tab. 4 initially evaluates the impact of altering the initialization of the baseline SparseBEV from grid-like (A) to radial (B), while retaining the common practice that generates several sampling points around each query. We find that radial initialization, without changing the sampling method, impairs efficient object-level feature extraction due to the sparse query

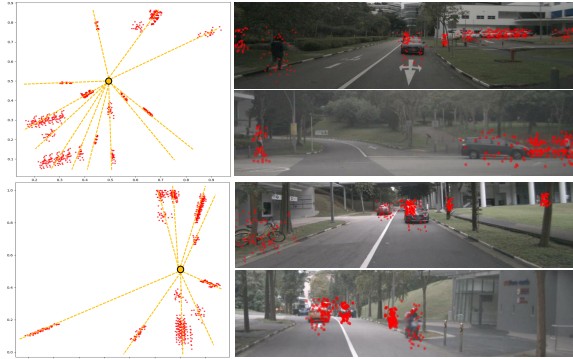

**Figure 6: Visualization of the ray sampling points. The sampling points with high prediction scores are marked with red dots, distributed in a radial pattern on the BEV (left) and in a vertical alignment in the image (right).**

**Table 5: Ablation studies about polar and rectangular rasterization for BEV features.**

| BEV Rasterization | mAP↑ | NDS↑ | mATE↓ |
|---|---|---|---|
| Polar | 0.340 | 0.402 | 0.726 |
| Rectangular | 0.345 | 0.406 | 0.717 |

distribution on each ray, resulting in reduced mAP and NDS. Comparing the result (C) with the baseline (A), even with an equivalent number of sampling points, image ray sampling under radial query initialization enhances mAP and NDS by 1.2% and 0.5% respectively. This demonstrates the critical importance of aligning the query initialization with the sampling pattern.

**BEV Ray Sampling:** As demonstrated in Tab. 4, we examine the influence of BEV ray sampling. When compared to sampling only on image features (C), integrating BEV feature sampling (D) leads to a 1.3% and 0.2% increase in mAP and NDS, and a 0.9% reduction in mATE. As such, it is demonstrated that dual-perspective feature sampling enhances object identification and classification. Tab. 5 elaborates on ablation studies for polar and rectangular rasterization of the generated BEV features. For rectangular rasterization, we set the number of BEV grids as $128 \times 128$, while for polar rasterization, it is set to $314 \times 80$. All other network structures and parameters remain consistent. Our results indicate that rectangular grids outperform polar grids, achieving a 0.5% and 0.4% increase in mAP and NDS, respectively.

**Foreground Query Supplement:** We present the results of 2D guided foreground query supplement in Tab. 4. To avoid ambiguity caused by the gains from the simultaneous 2D perception learning, we first compare the accuracy changes when solely using 2D object detection as an auxiliary learning task in (D) and (E), observing an increase of 0.2% in mAP and 0.8% in NDS. With the addition of foreground queries selected on the rays that intersect the predicted 2D bounding boxes (F), mAP and NDS further increase by 0.3% and 0.6%, respectively, while mATE decreases by 0.7%.

**Initialization and Sampling Configuration:** In Tab. 6, we simplify our RayFormer by excluding the 2D auxiliary task and 2D-guided query enhancement, focusing on the core performance. The

**Table 6: Ablations about the number of rays ($N_r$) and depth segments ($N_d$) on initialization and the number of adaptive ray points ($N_{arp}$) and sampling offsets per ray point ($N_{sf}$).**

| ID | $N_r$ | $N_d$ | $N_{arp}$ | $N_{sf}$ | mAP↑ | NDS↑ | mATE↓ |
|---|---|---|---|---|---|---|---|
| A | 120 | 6 | 3 | 4 | 0.337 | 0.397 | 0.723 |
| B | 150 | 6 | 3 | 4 | 0.345 | 0.406 | 0.717 |
| C | 180 | 6 | 3 | 4 | 0.352 | 0.416 | 0.708 |
| D | 150 | 3 | 3 | 4 | 0.336 | 0.396 | 0.721 |
| E | 150 | 6 | 5 | 4 | 0.343 | 0.408 | 0.716 |
| F | 150 | 6 | 3 | 1 | 0.334 | 0.395 | 0.729 |

**Table 7: Ablation study about the number of history frames. The training epoch is set to 24.**

| H | mAP↑ | NDS↑ | mATE↓ | mAVE↓ | mAOE↓ |
|---|---|---|---|---|---|
| 0 | 0.350 | 0.420 | 0.709 | 0.800 | 0.554 |
| 1 | 0.391 | 0.494 | 0.677 | 0.328 | 0.546 |
| 3 | 0.431 | 0.523 | 0.621 | 0.276 | 0.501 |
| 7 | 0.445 | 0.541 | 0.601 | 0.267 | 0.477 |

table shows how detection results vary with the number of rays and depth segments in query initialization. Increasing the number of rays improves performance, as seen in the progression from (A) to (C). Specifically, using 6 queries per ray (D) over 3 queries (B) increases mAP and NDS by 0.9% and 1.0%, respectively. Adjusting the adaptive ray points per query, as in (B) versus (E), has minimal impact. However, increasing the sampling offsets per ray point from 1 to 4, as in (B) versus (F), significantly enhances mAP and NDS by 1.1%. We select configuration (B) as the default for its optimal balance of precision and efficiency.

**Number of Historical Frames:** The impact of multi-frame input on detection performance is evident in Tab. 7. Adding a single historical frame to the key-frame input results in a notable increase in mAP by 4.1% and a significant boost in NDS by 7.4%, primarily due to a marked reduction in mAVE. Continued increase in the number of historical frames further reduces mAVE and mATE, highlighting the crucial contribution of sequential inputs to velocity estimation and the overall improvement in object detection accuracy.

## 5 Conclusion

In this paper, we fully consider the principles of camera optics and introduce RayFormer, which enhances multi-camera 3D object detection via ray-centric strategies. It initializes queries radially and selects sampling points on the associated radial segments to sample image features along with BEV features generated by the lift-splat-shoot method. Moreover, RayFormer presets varying ray densities on the BEV based on the categories and extends the predicted 2D bounding boxes to select foreground rays, providing additional queries close to the locations of actual objects. Extensive experiments on the nuScenes dataset with RayFormer show significant improvements over the baseline SparseBEV, highlighting the efficacy of ray-centric query initialization and feature sampling in enhancing multi-camera 3D object detection performance.

## Acknowledgments

This work is supported by the National Natural Science Foundation of China (No. 62332016).

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
