# OpenReview forum: "RayFormer: Improving Query-Based Multi-Camera 3D Object Detection via Ray-Centric Strategies"
_acmmm.org/ACMMM/2024/Conference — MM2024 Poster_

### Official Review · Reviewer_zEqJ · 2024-05-24

**Rating:** 4
**Confidence:** 3

**Summary:**

RayFormer is a novel 3D object detection model that improves the accuracy of multi-camera detection systems by aligning object queries with the optical characteristics of cameras. It introduces a unique approach by initializing queries along camera rays. This method reduces feature overlap and enhances distinctiveness, leading to improved detection performance as demonstrated on the nuScenes dataset, where RayFormer achieves significant improvements over existing benchmarks.

**Strengths:**

1. RayFormer is built on a solid rationale that addresses a key challenge in 3D object detection — the ambiguity of features among overlapping object queries in multi-camera systems. The model targets a fundamental problem that can significantly enhance detection accuracy by focusing on this issue.
2. The approach of RayFormer is logically structured, aligning the initialization of object queries with the optical characteristics of cameras.

**Limitations:**

1. While RayFormer introduces a ray-based query initialization as its core innovation, the performance improvements reported significantly rely on BEV sampling and initialization from 2D detection. This raises concerns about whether the primary contribution of ray-based querying is robust enough. By simply changing the distribution of queries from a Cartesian coordinate system to a polar coordinate system, and then querying image features for 3D detection like BEVFormer, how much improvement can be achieved?
2. RayFormer's method of initializing 3D detection queries from 2D detection results has been widely used in many methods[1]. This reduces the innovation aspect of the approach.

[1] Object as Query: Lifting Any 2D Object Detector to 3D Detection

**Suitability:**

2

---

### Official Review · Reviewer_GkmX · 2024-05-31

**Rating:** 4
**Confidence:** 3

**Summary:**

The paper introduces RayFormer, a 3D object detection framework that leverages ray-centric strategies to address feature ambiguity in multi-camera setups. The proposed method aligns query initialization and feature extraction with the optical characteristics of cameras, transforming perspective-view image features into bird's eye view (BEV) and segmenting the BEV map based on camera rays. The RayFormer is validated on the nuScenes dataset.

**Strengths:**

The paper proposes a novel approach by using a ray-centric query initialization and sampling strategy, which is well-aligned with the optical characteristics of cameras.

 The authors provide extensive experimental results on the nuScenes dataset, demonstrating the effectiveness of their method.

The paper is well-structured, and the methodology is clearly explained, making it easier for readers to follow.

**Limitations:**

While the ray-centric approach is innovative, the overall improvement in performance metrics (mAP and NDS) over the baseline SparseBEV is relatively modest (1.2% and 0.7%, respectively). The practical impact of these improvements should be more critically assessed.

The proposed method introduces significant complexity in the initialization and sampling process. The paper should provide a more detailed analysis of the computational overhead and its implications for real-time applications.

**Suitability:**

2

---

### Official Review · Reviewer_EPTK · 2024-06-01

**Rating:** 4
**Confidence:** 2

**Summary:**

RayFormer is a novel query-based multi-camera 3D object detector that improves accuracy by aligning query initialization and feature extraction with camera optical characteristics. It transforms image features into bird’s eye view (BEV) and segments the BEV map based on camera rays, enhancing object detection by extracting distinct features for each query. Evaluated on the nuScenes dataset, RayFormer achieves state-of-the-art performance, significantly surpassing previous methods.

**Strengths:**

1. Novel idea: The approach is innovative by selecting and defining queries from the perspective of the polar coordinate system and obtaining keys and values from BEV features.
2. Thorough experiments: The main experimental results clearly demonstrate the effectiveness of the method, while the ablation study highlights the effectiveness of all modules.

**Limitations:**

1. Writing clarity: The principles behind Query Initialization and Ray Sampling are not effectively illustrated with images, making the article difficult to understand.
2. Experimental results: Although the experimental results have shown significant improvement, they still lag behind the current leaderboard on nuScenes, including methods like BEVFormer v2 and VideoBEV.

**Suitability:**

1

---

### Meta-Review · Area_Chair_NvcZ · 2024-07-03

**Recommendation:** Accept (Poster)
**Confidence:** 4

**Metareview:**

The innovative approach uses the polar coordinate system to select and define queries and obtain keys and values from BEV features, which might inspire future research. Thorough experiments have been conducted, with the main results demonstrating the effectiveness of the method. The ablation study highlights the effectiveness of all modules involved in the approach. Despite these advantages, the approach itself is complicated in initialization, where computational overheads should be added to the final version.

Overall, the pros outweigh the cons. All reviewers also agree to accept this paper.  Thus, the AC suggests accepting this paper.